# Calibration-Aware Semi-Supervised Fetal Head Segmentation with Boundary-Positive Contrast

Ufaq Khan[*][1]                                    UFAQ.KHAN@MBZUAI.AC.AE
Umair Nawaz[1]                                   UMAIR.NAWAZ@MBZUAI.AC.AE
Tajamul Ashraf[1]                              TAJAMUL.ASHRAF@MBZUAI.AC.AE
Tausifa Jan Saleem[1]                         TAUSIFA.SALEEM@MBZUAI.AC.AE
Massimo Caputo[2]                                 M.CAPUTO@BRISTOL.AC.UK
Srinivas Ananth Narayan[3]                   SRINIVAS.NARAYAN@UHBW.NHS.UK
Muhammad Bilal[4]                              MUHAMMAD.BILAL@BCU.AC.UK
Junaid Qadir[5]                                       JQADIR@QU.EDU.QA
Muhammad Haris[1]                            MUHAMMAD.HARIS@MBZUAI.AC.AE

[1] *Mohamed Bin Zayed University of Artificial Intelligence, UAE*

[2] *University of Bristol, UK*

[3] *University Hospital Bristol and Weston, UK*

[4] *Birmingham City University, UK*

[5] *Qatar University, Qatar*

**Editors:** Accepted for publication at MIDL 2026

## Abstract

Accurate fetal head segmentation in ultrasound is hard to scale as labels are scarce and most errors occur at the head–background interface under speckle, shadowing, and low contrast. We present *UltraSemiNet*, a teacher–student framework that makes cross–pseudo supervision (CPS) *selective* via temperature calibration and a dual gate requiring high confidence and test-time augmentation (TTA) stability. We also introduce two boundary-focused modules that complement CPS: **SAT**, a boundary-positive spatial contrast that learns *through* ambiguous edges using an entropy belt and a soft-IoU agreement test; and **PCM**, a prototype-guided curriculum that maintains uncertainty-weighted head/background prototypes and targets feature–prototype discrepancies. Across two datasets (FBUI and HC18), UltraSemiNet improves overlap and boundary metrics over a calibrated CPS baseline (e.g., Dice $0.927{\rightarrow}0.971$; HD95 $7.9{\rightarrow}6.8$ px), with similar cross-dataset trends. Crucially, the calibrated gate reduces miscalibration of the *accepted* pseudo-labels: both expected calibration error (ECE) and Brier score decrease overall, with the largest gains within the 0–2 px boundary band, alongside improvements in pseudo-label accuracy. Ablations show CPS calibration, SAT, and PCM are complementary and concentrate improvements on boundary-sensitive metrics. In a blinded study, UltraSemiNet achieved better segmentation performance than two senior fetal medicine experts when evaluated against the dataset reference masks, indicating the potential to reduce manual refinements. Our code can be accessed on UltraSemiNet.

**Keywords:** Fetal Segmentation, Semi-Supervised Learning, Boundary-Aware Learning

---

[*] Corresponding Author

## 1. Introduction

The accurate segmentation of the fetal head on ultrasound is a foundational step for fetal biometry, in particular for head circumference (HC) and biparietal diameter (BPD) estimation. Clinically, HC and BPD are among the most routinely acquired fetal biometric measurements and are central to (i) pregnancy dating/gestational age (GA) estimation beyond the first trimester and (ii) longitudinal growth surveillance to detect growth disorders such as fetal growth restriction (FGR) and large-for-gestational-age, which are associated with adverse perinatal outcomes (Salomon et al., 2019; Figueras et al., 2018). These measurements are interpreted against widely used international standards (e.g., INTERGROWTH-21st and WHO), which model expected HC/BPD trajectories across gestation for clinical decision-making and cross-population comparability (Papageorghiou et al., 2014; Kiserud et al., 2017). Importantly, clinical practice guidelines emphasize standardized acquisition planes, visibility of key anatomical landmarks, and accurate ellipse/caliper placement under routine quality control (Salomon et al., 2019). In this context, a reliable head mask is not only a segmentation output, but it is a practical pre-measurement representation that stabilizes ellipse-based HC/BPD tools, reduces manual contour editing, and can support real-time quality control by flagging frames where the skull boundary is unreliable.

At the same time, the skull–background interface is precisely where ultrasound is most challenging due to speckle, acoustic shadowing, attenuation, and low local contrast (Noble and Boukerroui, 2006). These factors create two coupled obstacles: *(i)* label scarcity at scale (pixel-accurate annotation is time-consuming and requires domain expertise), and *(ii)* boundary ambiguity even when labels exist. The latter is compounded by inter-operator variation in annotation conventions (e.g., ellipse-like contours vs. fine-grained masks) (Zhang et al., 2020). As a consequence, fully supervised models can overfit limited labels or produce contours that appear plausible but are misaligned at the boundary, inflating boundary-sensitive errors such as Average Surface Distance (ASD) and the 95th percentile Hausdorff Distance (HD95) despite reasonable Dice scores (Shen et al., 2023).

Semi-supervised segmentation is therefore attractive because it can leverage large unlabeled collections through self-training and consistency regularization (Tarvainen and Valpola, 2017a; Chen et al., 2021a). However, in fetal ultrasound, the pixels most likely to be mislabeled are those near the head boundary. If pseudo-label selection relies on raw softmax confidence, which is often miscalibrated, self-training can systematically amplify edge errors. Similarly, enforcing consistency without checking whether predictions are stable under perturbations can also reinforce acquisition artifacts (Guo et al., 2017; Wang et al., 2019). A subtler issue is that many representation-learning strategies avoid uncertain regions when forming contrastive pairs (e.g., sampling only confident pixels or memory exemplars), which sharpens features in easy interiors while leaving boundary features under-constrained (Wang et al., 2022, 2021; Alonso et al., 2021). Together, miscalibrated pseudo-labels and boundary avoidance can create a gap between overlap metrics and the boundary fidelity required for reliable biometry.

We address semi-supervised binary fetal head segmentation in ultrasound with minimal pixel-level annotation. The technical challenge is to exploit unlabeled data without reinforcing boundary errors while learning representations that keep the head and background well separated, particularly at the interface. Concretely, we seek a training recipe that *(i)* selects

reliable pseudo-labels under explicit checks for confidence and augmentation stability, *(ii)* learns discriminative features through ambiguous boundary neighborhoods only when local evidence agrees, and *(iii)* regularizes the global feature space so that the two classes form compact, well-separated clusters despite class imbalance and annotation-style variability.

We propose *UltraSemiNet*, a teacher–student framework that combines cross–pseudo supervision (CPS) with two uncertainty-aware representation modules tailored to ultrasound boundaries. In practice, UltraSemiNet supports two complementary clinical roles: *(i)* as a pre-measurement step, it produces a clean head mask that stabilizes ellipse-based HC/BPD tools and reduces manual contour editing, and *(ii)* as a real-time quality check, its calibrated confidence and augmentation-stability scores can flag frames with unreliable boundaries for immediate reacquisition before measurements are finalized. Methodologically, we *(i)* calibrate teacher probabilities via temperature scaling on a small labeled subset and gate pseudo-labels using high binary confidence and stability under flip/rotation test-time augmentation (TTA); *(ii)* introduce **SAT**, a boundary-positive **SpAT**ial contrast that detects an entropy-based "boundary belt" and admits positive pairs across edges only when local probability fields agree (soft-IoU gate); and (iii) introduce **PCM**, a prototype-guided curriculum miner that maintains uncertainty-weighted teacher prototypes for head/background and prioritizes pixels whose student features are far from their predicted class prototype. Although we focus on 2D fetal head ultrasound in this work, the proposed calibration-aware selective supervision and boundary-focused representation learning are broadly applicable and can be extended to other obstetric targets and to 3D/volumetric acquisitions, as discussed in the limitation section.

**Contributions.***(i)* We formulate a *calibrated* CPS scheme for binary ultrasound segmentation that accepts pseudo-labels only when both confident and TTA-stable, reducing confirmation bias near boundaries. *(ii)* We propose **SAT**, an uncertainty-aware, boundary-positive spatial contrast that admits cross-edge positives via a soft-IoU agreement test on probability crops. *(iii)* We propose **PCM**, a lightweight prototype-guided curriculum that replaces heuristic hard-patch mining with uncertainty-weighted teacher prototypes and a discrepancy-based selection of semantically hard pixels, improving global feature separation under class imbalance. *(iv)* We conduct a blinded reader study with two senior fetal medicine experts, demonstrating higher agreement with UltraSemiNet outputs.

## 2. Related work

**Calibration-aware pseudo-labeling.** The utility of pseudo-labels depends on how well predicted probabilities reflect correctness. For this purpose, simple temperature scaling improves this alignment and yields cleaner supervision when applied before thresholding or re-labeling (Choi et al., 2024; Joy et al., 2023). Beyond a single global cutoff, adaptive acceptance schedules either class-balanced self-training or curriculum thresholds, reduce bias toward majority regions, and progressively admit harder pixels. Stability checks under flip/rotation test-time augmentation (TTA) further filter unreliable pseudo-labels (Tan et al., 2024) that would otherwise reinforce boundary artifacts. In medical SSL, uncertainty-aware teachers reduce the weight of the ambiguous regions or gate them to curb error propagation near the interfaces (Alizadehsani et al., 2024; Vazhentsev et al., 2025; Chen et al., 2023).

UltraSemiNet adopts this calibration-aware view end-to-end, where we (i) temperature-scale the teacher on a small labeled split each epoch, (ii) enforce a dual gate that requires both high *calibrated* confidence and TTA-based stability before a pixel supervises CPS, and (iii) reuse the same calibrated probabilities to drive SAT's boundary-positive pairing and PCM's uncertainty-weighted prototype updates, ensuring selection and representation shaping are governed by a shared reliability criterion.

**Classical Techniques in Fetal Ultrasound Segmentation.** The segmentation of fetal structures in ultrasound images initially relied heavily on classical image processing methods. The techniques such as thresholding (Liu et al., 2019), region growth (Yuheng and Hao, 2017), and Active Contour Models (ACM) (Kass et al., 1988) formed the basis of early segmentation efforts. While these methods provided initial frameworks for segmentation, they often required substantial manual intervention and struggled with the intrinsic challenges of ultrasound imaging, such as speckle noise and low contrast(Khan et al., 2024), limiting their effectiveness and reliability in clinical applications (Smith and Doe, 2010; Khan et al., 2026). The advent of deep-learning architectures has substantially advanced fetal ultrasound segmentation.

## 3. Methodology

**Overview.** We present the method in the same order it is executed during training: (i) update the exponential moving average (EMA) teacher, calibrate its probabilities, and select reliable pseudo-labels via a confidence+stability gate; (ii) use the calibrated teacher to define boundary-aware positive/negative pairs and apply spatial contrast; (iii) maintain uncertainty-weighted class prototypes and perform curriculum mining to regularize the global embedding space.

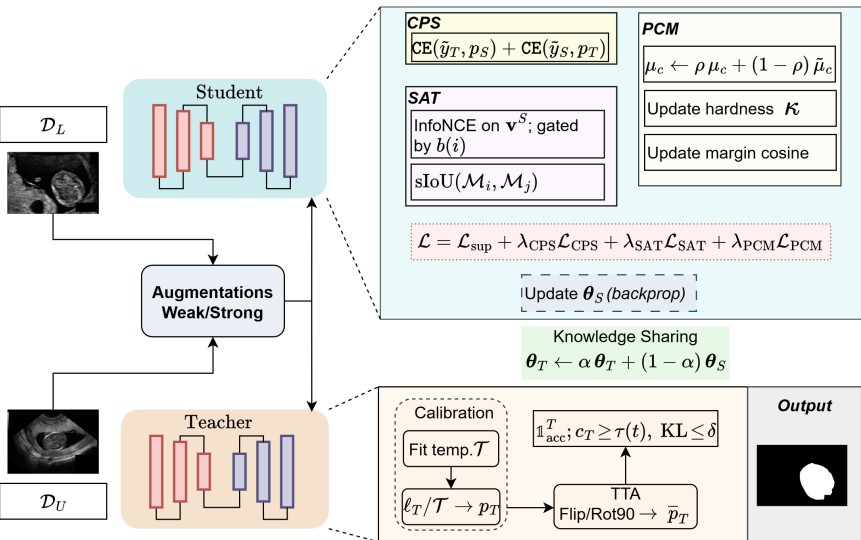

Figure 1: An overview of UltraSemiNet: weak/strong views feed teacher $f_T$ (temperature-calibrated, TTA-averaged) and student $f_S$. The student minimizes the overall loss $\mathcal{L}$ using CPS, boundary-positive SAT, and PCM modules, while the teacher is updated via EMA.

### 3.1. Preliminaries and Notation

Let $\mathcal{D}_L = \{(x, y)\}$ be labeled images with binary masks $y \in \{0, 1\}^{h \times w}$ where (1=head, 0=background), and $\mathcal{D}_U = \{x\}$ unlabeled images as depicted in Fig. 1. We use a student network $f_S(\cdot; \theta_S)$ and an EMA teacher $f_T(\cdot; \theta_T)$ which can be updated as:

$$\theta_T \leftarrow \alpha\, \theta_T + (1 - \alpha)\, \theta_S, \qquad \alpha \in [0.99, 0.999]. \tag{1}$$

At pixel $i$, the networks output head probabilities $p_S(i), p_T(i) \in [0, 1]$ and the teacher pseudo-label $\hat{y}_T(i) = \mathbb{1}[p_T(i) \geq 0.5]$. The teacher uncertainty is measured by binary entropy as:

$$u(i) = -p_T(i) \log p_T(i) - (1 - p_T(i)) \log(1 - p_T(i)), \tag{2}$$

and we use normalized entropy $\tilde{u}(i) = u(i)/\log 2 \in [0, 1]$ to identify boundary regions. For representation learning, we use $\ell_2$-normalized pixel embeddings $v_i \in \mathbb{R}^d$ from a lightweight projection head.

### 3.2. Calibrated Cross–Pseudo Supervision (CPS)

CPS leverages unlabeled data by letting teacher and student supervise each other. However, using raw probabilities directly can therefore risk reinforcing errors. We therefore calibrate teacher probabilities using calibration techniques (Kim et al., 2025; Forest and Fink, 2024) so that confidence aligns better with empirical correctness, and we restrict CPS to pixels that are both confident and stable under simple test-time augmentations.

**Probability calibration.** At the start of each epoch, we fit a temperature $\mathcal{T} > 0$ on a small labeled subset by minimizing negative log-likelihood. If $\ell_T(i)$ denotes the teacher head logit, the *calibrated* probability is $p_T(i) \leftarrow \sigma(\ell_T(i)/\mathcal{T})$, where, $\sigma(z) = \frac{1}{1+e^{-z}}$. Calibration is re-estimated periodically (e.g., every 3 epochs) to track the evolving teacher. All downstream decisions in the epoch use the calibrated $p_T$. For binary segmentation, a natural confidence is $c_T(i) = \max\{p_T(i), 1 - p_T(i)\}$. A pixel is accepted for CPS only if it is (i) confident and (ii) stable under simple TTA (flips/rot90). Let $\bar{p}_T^{\text{TTA}}(i)$ be the mean teacher probability under these augmentations (mapped back). We measure stability using the Bernoulli KL divergence between $p_T(i)$ and $\bar{p}_T^{\text{TTA}}(i)$:

$$\mathbb{1}_{\text{acc}}^T(i) = \mathbb{1}[c_T(i) \geq \tau(t)] \times \mathbb{1}\left[\text{KL}\left(\text{Bern}(p_T(i)) \,\|\, \text{Bern}(\bar{p}_T^{\text{TTA}}(i))\right) \leq \delta\right] \tag{3}$$

where $\tau(t)$ anneals linearly from 0.95 to 0.80 over training and $\delta$ is a small KL cutoff. Now, the accepted teacher labels are:

$$\tilde{y}_T(i) = \begin{cases} \hat{y}_T(i), & \text{if } \mathbb{1}_{\text{acc}}^T(i) = 1, \\ \text{ignore}, & \text{otherwise.} \end{cases} \tag{4}$$

Symmetrically, we obtain $\tilde{y}_S(i)$ with $p_S(i)$ and $c_S(i) = \max\{p_S(i), 1 - p_S(i)\}$, defining $\mathbb{1}_{\text{acc}}^S(i)$ via (3). Here, $\hat{y}_T(i)$ denotes the *raw teacher pseudo-label* at pixel $i$, obtained from the calibrated probability $p_T(i)$ whereas $\tilde{y}_T(i)$ represent the *accepted* pseudo-label. We then

supervise the student using teacher-accepted pixels $\Omega_T = \{i : \mathbb{1}_{\mathrm{acc}}^T(i) = 1\}$, and supervise the teacher (EMA) using student-accepted pixels $\Omega_S = \{i : \mathbb{1}_{\mathrm{acc}}^S(i) = 1\}$:

$$\mathcal{L}_{\mathrm{CPS}} = \frac{1}{|\Omega_T|} \sum_{i \in \Omega_T} \mathrm{CE}\big(\tilde{y}_T(i), p_S(i)\big) + \frac{1}{|\Omega_S|} \sum_{i \in \Omega_S} \mathrm{CE}\big(\tilde{y}_S(i), p_T(i)\big). \tag{5}$$

By restricting CPS to confident and augmentation-stable pixels, we reduce confirmation bias where the model is most error-prone (near the head boundary) and provides consistent, calibrated probabilities $p_T(i)$ and uncertainty $u(i)$ as depicted in Figure 4(b) of supplementary material.

### 3.3. SAT: Uncertainty-Aware Boundary-Positive Spatial Contrast

The SAT module is designed around a simple observation, i.e., in binary fetal head segmentation, most of the ambiguity lies exactly where the head meets the background. Rather than discarding these pixels as "noisy," SAT turns them into useful supervision, but only when the nearby evidence supports doing so. In practice, SAT uses the *calibrated* head probabilities of the teacher $p_T(i) \in [0, 1]$ (Sec. 3.2) to identify potentially ambiguous locations and then shapes the student's representations to be consistent across those locations and well-separated from the background.

We begin by quantifying local uncertainty using the binary entropy: $u(i)$, and normalize it by its maximum, $\tilde{u}(i)$. Pixels with intermediate $\tilde{u}$ are likely to lie on (or near) the head boundary. We therefore form a *boundary belt* by marking pixels whose normalized entropy falls in $[\epsilon_1, \epsilon_2]$, so $b(i) = \mathbb{1}\big[\epsilon_1 \leq \tilde{u}(i) \leq \epsilon_2\big]$, where $\epsilon_1 = 0.40$, $\epsilon_2 = 0.95$.

Now, ambiguous pixels are not automatically trusted. Before we let them act as "positives" in a contrastive pair, we ask whether the local probability fields around two candidate pixels agree. For that we extract a single-channel probability crop $\mathcal{M}_i \in [0, 1]^{s \times s}$ centered at $i$ and compute a soft intersection-over-union with a neighbor $j$,

$$\mathrm{sIoU}(\mathcal{M}_i, \mathcal{M}_j) = \frac{\sum_\omega \min\{\mathcal{M}_i(\omega), \mathcal{M}_j(\omega)\}}{\sum_\omega \max\{\mathcal{M}_i(\omega), \mathcal{M}_j(\omega)\}}, \tag{6}$$

which measures how similar the surrounding head probabilities are within an $s \times s$ window.

**Positive/negative sampling with CPS consistency.** With these ingredients, SAT constructs contrastive pairs as follows. For each anchor pixel $i$, we sample one positive within a small radius $r$ that shares the same pseudo-label, and $K_-$ negatives from a wider ring region whose pseudo-label is opposite. If the anchor is away from the boundary ($b(i) = 0$), we simply require that both pixels are eligible under the CPS acceptance rule (Eq. (3)), ensuring that they are confident and stable. If the anchor lies in the boundary belt ($b(i) = 1$), we only admit a positive with probability $q_b(t)$ and when the local fields agree, i.e., $\mathrm{sIoU}(\mathcal{M}_i, \mathcal{M}_{p(i)}) > \tau_{\mathrm{bIoU}}$. Moreover, at least one of the two pixels must be CPS-accepted to guard against spurious matches. Now, the negatives are drawn from an annulus and must carry the opposite pseudo-label. Here, to avoid background overwhelming the loss in this binary setting, we sample negatives so that head and background contribute in a balanced way whenever possible.

To focus learning where it matters, each anchor is given a weight that increases with local ambiguity but does not reward extreme uncertainty. Mathematically, it can be shown

as: $w(i) = \left( \gamma_1 \left( 1 - \tilde{u}(i) \right) + \gamma_2 \, \overline{\tilde{u}}_{\mathcal{N}_r(i)} \right) \left( 1 + \lambda_b \, b(i) \right)$, where $\overline{\tilde{u}}_{\mathcal{N}_r(i)}$ averages $\tilde{u}$ in the positive radius, $\gamma_1{=}0.7$, $\gamma_2{=}0.3$, and a small boost $\lambda_b{=}0.5$ nudges attention toward the belt. Using $\mathcal{L}_2$-normalized student embeddings, SAT minimizes a cosine InfoNCE loss as:

$$\mathcal{L}_{\mathrm{SAT}} = \sum_i w(i) \left[ -\log \frac{\exp\left( \langle v_i^S, v_{p(i)}^S \rangle / \beta \right)}{\exp\left( \langle v_i^S, v_{p(i)}^S \rangle / \beta \right) + \sum_{n \in \mathcal{N}(i)} \exp\left( \langle v_i^S, v_n^S \rangle / \beta \right)} \right], \quad \beta = 0.07 \tag{7}$$

In effect, SAT asks the representation to be continuous across genuinely consistent boundary neighborhoods and separate them decisively from the background nearby. Now, as admission and weighting depend on the same calibrated probabilities and acceptance rule used in CPS, the contrastive signal remains logically consistent with supervision selection, i.e, ambiguous regions shape the features only when the evidence supports it, leading to sharper, more reliable contours (improved HD95/ASD) without sacrificing Dice. In practice, we use $r{=}5$, $s{=}15$, $K_-{=}64$ negatives per anchor, ramp $q_b(t)$ from 0 to 0.4 during the first half of training, and choose $\tau_{\mathrm{bIoU}} \in \{0.5, 0.6, 0.7\}$.

### 3.4. PCM: Prototype-Guided Curriculum Miner

Where SAT shapes features *locally* around the boundary, PCM provides a lightweight *global* structure by maintaining running prototypes for "head" and "background" and asks the student's features to organize themselves around these references. This is done in a way that is robust to label noise, i.e., prototypes are updated only from pixels that the teacher deems reliable under the CPS acceptance rule, and those contributions are further down-weighted when uncertainty is high.

Specifically, for each class $c \in \{0, 1\}$ we maintain an EMA prototype $\mu_c$. At a training step, teacher embeddings $v_i^T$ with pseudo-label $\hat{y}_T(i) = c$ are pooled into a temporary estimate $\tilde{\mu}_c$ using weights: $\omega(i) = \mathbb{1}_{\mathrm{acc}}^T(i) \left( 1 - \tilde{u}(i) \right)$, so that only confident, augmentation-stable pixels contribute and highly uncertain cases have little influence. The class prototype is then updated by exponential averaging,

$$\tilde{\mu}_c = \frac{\sum_{i: \hat{y}_T(i)=c} \omega(i) \, v_i^T}{\sum_{i: \hat{y}_T(i)=c} \omega(i)}, \quad \mu_c \leftarrow \rho \, \mu_c + (1 - \rho) \, \tilde{\mu}_c, \quad \rho = 0.99. \tag{8}$$

We warm-start $\mu_c$ from the labeled set. If a batch contains no reliable pixels for a class, the EMA simply carries the previous prototype forward. Not all pixels are equally informative for organizing the feature space. To determine where PCM should act most strongly, we define a semantic hardness score at each pixel by measuring how far the student's feature is from the prototype of its own predicted class. Let $p_S(i)$ be the student head probability (so $1 - p_S(i)$ is background). We compute $\kappa(i) = \max\left\{ p_S(i) \left( 1 - \cos\langle v_i^S, \mu_1 \rangle \right), (1 - p_S(i)) \left( 1 - \cos\langle v_i^S, \mu_0 \rangle \right) \right\}$, which is large when a pixel is predicted as head (or background) but its feature is inconsistent with the corresponding prototype. A curriculum then selects the top $\gamma(t)$ fraction of pixels by $\kappa(i)$ and forms the active set $\Omega_\gamma$. Because background typically occupies more area than head, we subsample background within $\Omega_\gamma$ so that both classes contribute comparably.

Finally, PCM pulls features toward the prototype of their predicted class while pushing them away from the other class, using a marginized cosine loss:

$$\mathcal{L}_{\text{PCM}} = \frac{1}{|\Omega_\gamma|} \sum_{i \in \Omega_\gamma} \left[ \pi_1(i) \left( m - \cos\langle v_i^S, \mu_1 \rangle + \cos\langle v_i^S, \mu_0 \rangle \right)_+ + \pi_0(i) \left( m - \cos\langle v_i^S, \mu_0 \rangle + \cos\langle v_i^S, \mu_1 \rangle \right)_+ \right], m = 0.2. \tag{9}$$

with $(\cdot)_+ = \max(0, \cdot)$, the effect is to produce compact, well-separated head and background clusters in the embedding space, guided by prototypes that are themselves estimated from reliable pixels. As prototype updates use the same acceptance indicator and entropy as CPS, PCM's global organization is aligned with the supervision that trains the classifier head and with the local boundary shaping performed by SAT. In practice, we compute $\kappa(i)$ at resolutions of 1/2 and 1/4 for efficiency, then upsample the resulting mask. Here, if prototypes begin to drift or collapse, increasing the margin to $m=0.3$ or slowing the curriculum ramp mitigates the issue.

## 4. Experimental Setup

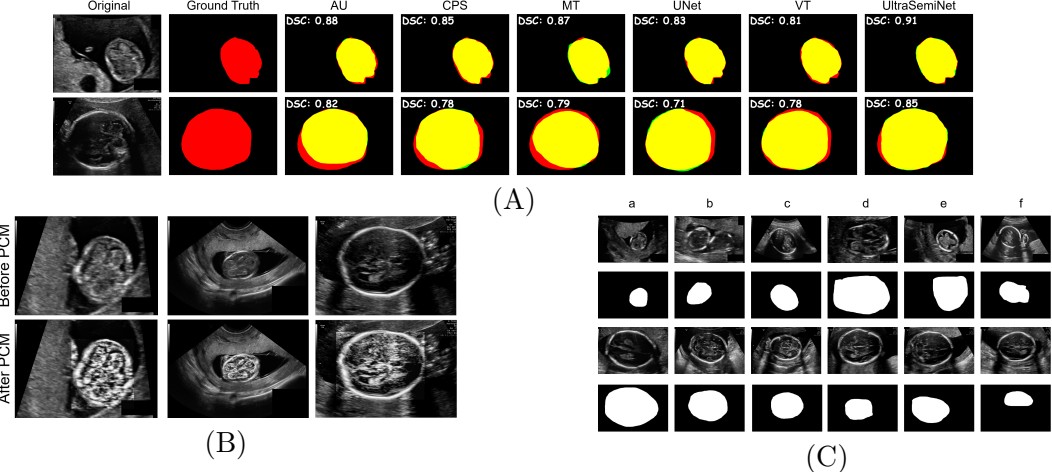

Figure 2: (A) Original ultrasound images, ground truth segmentations, and model outputs with Dice scores; (B) Before and after PCM; (C) Best (a-c) and worst (d-f) UltraSemiNet predictions.

**Datasets.** We evaluate on two fetal head ultrasound collections. (1) *FBUI* (Alzubaidi et al., 2023): 3,832 images spanning 18–40 weeks GA, split **by subject** into 60%/20%/20% train/val/test. (2) *HC18 subset* (van den Heuvel et al., 2018): It contains 999 fetal brain images with binary head annotations. Unless stated, FBUI provides labeled supervision, and contributes additional unlabeled images for semi-supervised training, while HC18 is used solely for cross-dataset evaluation. Further details are provided in the supplementary material.

**Cross-validation and cross-dataset tests.** All in-distribution results are reported under **five-fold** patient-level CV on FBUI. For cross-dataset generalization, models are trained on FBUI and evaluated on the HC18 subset without retuning thresholds.

**Evaluation metrics.** We report Dice Similarity Coefficient (DSC), Average Surface Distance (ASD), and 95th percentile Hausdorff distance (HD95). DSC measures volumetric

Table 1: **FBUI (in-distribution) results** under 5-fold cross-validation.

| Method | DSC ↑ | ASD (px) ↓ | HD95 (px) ↓ |
|---|---|---|---|
| U–Net (Ronneberger et al., 2015) | $0.913 \pm 0.015$ | $2.41 \pm 1.25$ | $9.8 \pm 4.2$ |
| Attention U–Net (Oktay et al., 2018) | $0.921 \pm 0.013$ | $2.12 \pm 1.19$ | $8.7 \pm 3.9$ |
| ViT (Dosovitskiy et al., 2020) | $0.906 \pm 0.018$ | $2.78 \pm 1.43$ | $11.3 \pm 5.1$ |
| 2D nnU–Net (Isensee et al., 2021) | $0.930 \pm 0.011$ | $1.94 \pm 1.07$ | $8.2 \pm 3.6$ |
| Mean Teacher (MT) (Tarvainen and Valpola, 2017b) | $0.918 \pm 0.014$ | $2.23 \pm 1.18$ | $9.3 \pm 4.0$ |
| CPS-only (Chen et al., 2021b) | $0.927 \pm 0.012$ | $1.86 \pm 1.09$ | $7.9 \pm 3.8$ |
| MedSAM (Ma et al., 2024) | $0.821 \pm 0.061$ | $6.85 \pm 3.72$ | $26.4 \pm 11.7$ |
| SAMUS (Lin et al., 2024) | $0.846 \pm 0.049$ | $5.72 \pm 3.11$ | $22.7 \pm 10.5$ |
| Cross-Teaching (Luo et al., 2022) | $0.944 \pm 0.046$ | $2.62 \pm 1.21$ | $8.2 \pm 4.9$ |
| MC-Net+ (Wu et al., 2022) | $0.884 \pm 0.042$ | $4.02 \pm 1.95$ | $10.2 \pm 3.5$ |
| **UltraSemiNet (ours)** | $0.971 \pm 0.010$ | $1.07 \pm 0.92$ | $6.8 \pm 3.2$ |

Table 2: **Cross-dataset results:** train on FBUI, test on HC18 (no retuning). We report ASD and HD95 in both pixels and millimeters using the provided per-image pixel spacing.

| Method | DSC ↑ | ASD ↓ | | HD95 ↓ | |
|---|---|---|---|---|---|
| | | px | mm | px | mm |
| U–Net (Ronneberger et al., 2015) | $0.887 \pm 0.028$ | $3.21 \pm 1.92$ | $0.45 \pm 0.27$ | $14.2 \pm 6.3$ | $1.99 \pm 0.88$ |
| Attention U–Net (Oktay et al., 2018) | $0.896 \pm 0.026$ | $2.98 \pm 1.78$ | $0.42 \pm 0.25$ | $13.1 \pm 5.8$ | $1.83 \pm 0.81$ |
| 2D nnU–Net (Isensee et al., 2021) | $0.905 \pm 0.022$ | $2.75 \pm 1.63$ | $0.38 \pm 0.23$ | $12.4 \pm 5.5$ | $1.73 \pm 0.77$ |
| Mean Teacher (MT) (Tarvainen and Valpola, 2017b) | $0.898 \pm 0.025$ | $2.91 \pm 1.71$ | $0.41 \pm 0.24$ | $12.8 \pm 5.7$ | $1.79 \pm 0.80$ |
| CPS-only (Chen et al., 2021b) | $0.907 \pm 0.022$ | $2.63 \pm 1.56$ | $0.37 \pm 0.22$ | $11.7 \pm 5.1$ | $1.64 \pm 0.71$ |
| MedSAM (Ma et al., 2024) | $0.803 \pm 0.072$ | $7.41 \pm 4.10$ | $1.04 \pm 0.57$ | $28.9 \pm 12.6$ | $4.04 \pm 1.76$ |
| SAMUS (Lin et al., 2024) | $0.828 \pm 0.064$ | $6.32 \pm 3.65$ | $0.88 \pm 0.51$ | $24.7 \pm 11.2$ | $3.45 \pm 1.57$ |
| Cross-Teaching (Luo et al., 2022) | $0.897 \pm 0.037$ | $2.98 \pm 2.76$ | $0.42 \pm 0.39$ | $11.5 \pm 6.8$ | $1.61 \pm 0.95$ |
| MC-Net+ (Wu et al., 2022) | $0.850 \pm 0.032$ | $4.17 \pm 2.56$ | $0.58 \pm 0.36$ | $13.9 \pm 9.4$ | $1.94 \pm 1.31$ |
| **UltraSemiNet (ours)** | $0.925 \pm 0.019$ | $1.27 \pm 1.44$ | $0.18 \pm 0.20$ | $10.3 \pm 4.7$ | $1.44 \pm 0.66$ |

overlap, ASD summarizes average contour deviation, and HD95 reflects worst-case boundary error while being robust to outliers. All metrics are computed per image and averaged over the test set.

**Baselines.** We compare *UltraSemiNet* against: U-Net (Ronneberger et al., 2015), Attention U-Net (Oktay et al., 2018), Vision Transformer (ViT) (Dosovitskiy et al., 2020), Mean Teacher (MT) (Tarvainen and Valpola, 2017b), and CPS-only (Chen et al., 2021b). To contextualize results with widely used references, we also include 2D nnU-Net (Isensee et al., 2021) (default config) and promptable foundation models MedSAM (Ma et al., 2024) and SAMUS (Lin et al., 2024).

**Semi-supervised protocol.** In each fold, we sample **1,500 labeled** images from the FBUI training split; the remaining FBUI training images are treated as **unlabeled**. The teacher–student model uses *calibrated* teacher probabilities (temperature refit every 3–5 epochs) and accepts unlabeled pixels for CPS only if they pass a binary confidence threshold annealed $0.95 \rightarrow 0.80$ and a flip/rot90 TTA stability test (KL $\leq 0.15$) (Sec. 3.2). For temperature scaling, we construct a small calibration set by randomly sampling $N=200$ labeled images from the training split of the current fold. The calibration set is strictly

disjoint from the validation and test subjects. To prevent leakage, our k-fold splitting is performed at the patient level, i.e., images are grouped by patient ID prior to partitioning, and all samples from the same patient are assigned to the same fold.

**Foundation-model protocol.** We include MedSAM (Ma et al., 2024) and SAMUS (Lin et al., 2024) as baselines for the foundation model and evaluate them using a consistent fine-tuning protocol on FBUI. Specifically, we fine-tune each model only on the FBUI-labeled split (no HC18 usage), using the same input resolution ($224 \times 224$), preprocessing/augmentations, and a comparable training budget. Following their prompt-based design, we use box prompts constructed as the tight bounding box around the head region from the training masks (with a fixed padding). All predictions are mapped back to the evaluation resolution before computing metrics.

**Implementation details.** Images are resized to $224\times224$ and intensity-normalized. The teacher uses *weak* views (flips, 90° rotations), whereas the student uses weak+strong views (brightness/contrast/gamma jitter, mild elastic, light speckle). Models are trained on $1\times$A100 with total batch size 32 for 10,000 iterations (five-fold CV unless noted). We use AdamW (lr $1\times10^{-4}$, weight decay $1\times10^{-4}$), cosine decay with 5% warmup, gradient clipping at 1.0. The student is optimized with $\mathcal{L}_{\text{sup}}+\lambda_{\text{CPS}}\mathcal{L}_{\text{CPS}}+\lambda_{\text{SAT}}\mathcal{L}_{\text{SAT}}+\lambda_{\text{PCM}}\mathcal{L}_{\text{PCM}}$ (defaults: $\lambda_{\text{CPS}}=1.0$, $\lambda_{\text{SAT}}=0.3$, $\lambda_{\text{PCM}}=0.2$). The teacher is updated by EMA with $\alpha=0.996$. SAT uses one positive and $K_- = 64$ negatives per anchor (radius $r=5$, annulus $[7, 11]$, gate $\tau_{\text{bIoU}}=0.6$), whereas PCM maintains head/background prototypes with EMA $\rho=0.99$ and a curriculum keep ratio $\gamma(t) : 0.3\rightarrow0.7$. Early stopping was also applied on validation DSC with patience of 30.

**Computational cost.** UltraSemiNet contains 17.27 million parameters and requires approximately 61.20 GFLOPs for a standard $1 \times 224 \times 224$ input. On a single NVIDIA A100 GPU, training takes approximately 55 seconds per epoch. At inference time, UltraSemiNet runs in 4.25 ms per sample (corresponding to $\sim$220 FPS), indicating that the proposed method can operate efficiently in real-time clinical workflows.

## 5. Results

**Main Results on FBUI (5-fold CV).** Table 1 reports results on FBUI under five-fold, patient-level cross-validation with 1,500 labeled images and the remaining training images treated as unlabeled. UltraSemiNet achieves the best Dice while yielding the largest gains on boundary-sensitive metrics (ASD, HD95). Improvements over CPS-only are modest in Dice (as expected for a relatively clear binary target) but consistent and more pronounced for HD95, indicating sharper and more reliable contours.

**Cross-Dataset Generalization (Train on FBUI, Test on HC18 subset).** We next assess robustness under distribution shift by training on FBUI and evaluating on the HC18 subset without retuning thresholds. All methods degrade, but UltraSemiNet maintains the strongest performance and the largest boundary gains (Table 2). Reporting ASD/HD95 in millimeters makes the results directly clinically interpretable and confirms that the improvements persist beyond resized pixel-space evaluation.

**Qualitative Analysis.** Figure 2A shows qualitative results along with a DSC score for each method where UltraSemiNet reduces boundary leakage. Figure 2B shows that PCM further refines contrast in high-entropy areas, whereas Figure 2C shows the best and worst

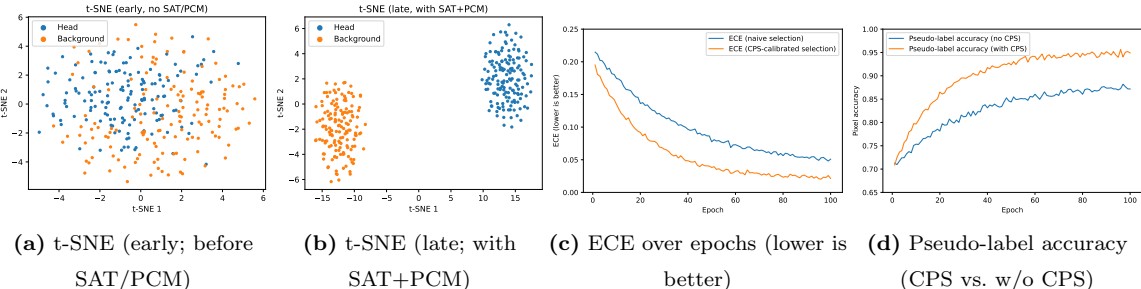

**(a)** t-SNE (early; before SAT/PCM)  **(b)** t-SNE (late; with SAT+PCM)  **(c)** ECE over epochs (lower is better)  **(d)** Pseudo-label accuracy (CPS vs. w/o CPS)

Figure 3: **UltraSemiNet dynamics.** (a–b) SAT+PCM progressively sharpen feature separation in t-SNE (more compact and separable head/background clusters). (c) Calibrated CPS with TTA-stability reduces expected calibration error (ECE) compared to naive selection. (d) CPS yields higher pseudo-label accuracy and faster convergence.

predictions for our model. Figure 3 summarizes why UltraSemiNet improves downstream accuracy. Here, SAT+PCM turns entangled early features into compact, separated clusters (a–b). CPS reduces ECE across training (c) and produces higher pseudo-label accuracy throughout (d). We also show some further analysis in Figure 4 of the appendix by presenting **dynamics** where CPS acceptance rises while TTA KL falls (a), boundary pixels steadily catch up to interiors as SAT focuses learning at ambiguous edges (b), and near-diagonal reliability at late epochs (c).

**Clinical and Practical Implications.** Table 3a compares UltraSemiNet predictions with two medical experts (ME1, ME2) on total 60 samples from the FBUI and HC18 datasets, revealing that UltraSemiNet consistently outperforms medical experts. In terms of role and experience, ME1 is a board-certified fetal medicine specialist with over 10 years of experience in obstetric ultrasound, while ME2 is a senior sonographer with over 13 years of dedicated practice in high-risk pregnancy imaging. Each expert independently segmented the fetal head boundary on the set of 30 images from each dataset. These annotations were then compared with the ground-truth masks provided by the source dataset and the predicted masks generated by our UltraSemiNet model. UltraSemiNet shows greater agreement with the consensus reference than either individual expert, suggesting that its contours are at least as consistent with the consolidated clinical standard as those of human raters in this setting.

**Ablations and Calibration** We isolate the contributions of calibration/gating, SAT, and PCM as depicted in Table 3b. All ablations use the same backbone, schedules, and labeled fraction. We quantify probability reliability using Expected Calibration Error (ECE; 15 bins) and Brier score on the student's probability map. Table 3c shows that calibrating CPS and using SAT/PCM reduces ECE and Brier versus naive CPS.

## 6. Limitations and Future Work

This paper targets 2D *binary* skull segmentation, which can be extended beyond the head to additional obstetric targets (e.g., abdominal circumference, femur length, placenta, and standard cardiac views), to multi-class anatomical segmentation, and to 3D/volumetric acquisitions, which is a natural next step. Concretely, for 3D/4D volumes, we would re-

Table 3: Comparative performance, ablations, and probability calibration. HD95/ASD in pixels (px).

| Dataset | UltraSemiNet (Ours) | | | Medical Expert 1 | | | Medical Expert 2 | | |
|---|---|---|---|---|---|---|---|---|---|
| | DSC ↑ | HD95 ↓ | ASD ↓ | DSC ↑ | HD95 ↓ | ASD ↓ | DSC ↑ | HD95 ↓ | ASD ↓ |
| **FBUI** | 0.971 | 6.800 | 1.070 | 0.965 | 14.160 | 1.160 | 0.949 | 23.020 | 1.230 |
| **HC18** | 0.925 | 10.300 | 1.270 | 0.919 | 17.450 | 1.530 | 0.897 | 26.270 | 3.870 |

a) Comparison with two senior medical experts.

| Variant | DSC ↑ | ASD ↓ | HD95 ↓ |
|---|---|---|---|
| CPS (no calib., no TTA) | $0.940 \pm 0.013$ | $1.90 \pm 1.10$ | $8.6 \pm 4.1$ |
| CPS (calib. & TTA gate) | $0.952 \pm 0.012$ | $1.55 \pm 1.05$ | $7.9 \pm 3.8$ |
| CPS + SAT (w/o PCM) | $0.962 \pm 0.011$ | $1.28 \pm 0.98$ | $7.2 \pm 3.5$ |
| CPS + PCM (w/o SAT) | $0.958 \pm 0.011$ | $1.34 \pm 1.00$ | $7.4 \pm 3.6$ |
| **UltraSemiNet (Full)** | $0.971 \pm 0.010$ | $1.07 \pm 0.92$ | $6.8 \pm 3.2$ |

b) Ablations on FBUI.

| Variant | ECE (%) ↓ | Brier ($\times 10^{-2}$) ↓ |
|---|---|---|
| CPS (no calibration) | $5.9 \pm 1.1$ | $3.8 \pm 0.4$ |
| CPS (calibrated) | $3.7 \pm 0.9$ | $3.5 \pm 0.3$ |
| **UltraSemiNet** | $\mathbf{2.9 \pm 0.8}$ | $\mathbf{3.3 \pm 0.3}$ |

c) Calibration on FBUI.

place the 2D backbone with a standard 3D encoder–decoder (e.g., 3D U-Net/nnU-Net-style 3D configuration (Çiçek et al., 2016; Isensee et al., 2021)), apply the same temperature calibration and confidence+stability gating voxel-wise using a small set of 3D-consistent flips/rotations, and lift SAT/PCM by defining the entropy belt, probability-crop sIoU check, and prototype mining over 3D patches/neighborhoods. The added cost scales mainly with volume size and the number of TTA views and can be controlled through patch-based training and low-resolution gating.

Although we observed cross-dataset gains, broader evaluation under stronger domain shifts (scanner vendors, protocols) and prospective studies are needed. Finally, integrating calibrated uncertainty with measurement tools (e.g., auto-ellipse placement), conformal risk control, and active selection for targeted annotation could further improve safety and data efficiency. Moreover, we note that SynthStrip (Hoopes et al., 2022) is a brain-extraction ("skull-stripping") method designed for 3D neuroimaging (MRI/CT) and is therefore not directly comparable to our task of 2D fetal ultrasound head boundary segmentation. We therefore restrict comparisons to methods applicable to 2D ultrasound segmentation under the same training protocol. As future work, we will investigate whether ultrasound-specific preprocessing or anatomy-prior approaches inspired by such methods can further improve robustness under severe attenuation and incomplete skull visibility.

## 7. Conclusion

We presented UltraSemiNet, a semi-supervised framework for fetal head ultrasound that combines calibrated cross-pseudo supervision with two boundary-aware modules, leveraging SAT and PCM. SAT heightens spatial sensitivity along ambiguous skull–background interfaces, while PCM organizes features globally via uncertainty-weighted class prototypes, together yielding sharper, more reliable contours. On FBUI (5-fold CV), UltraSemiNet reached 0.971 Dice and reduced HD95 to 6.8 px, with consistent boundary-metric gains on cross-dataset HC18 evaluations.

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

## Appendix A. Supplementary Material

### A.1. Overall Objective and Training

UltraSemiNet combines supervised learning on labeled images with three unlabeled objectives that are all derived from the *same, calibrated* teacher probabilities (Sec. 3.2). On labeled data $(x, y) \in \mathcal{D}_L$ (binary head mask $y \in \{0, 1\}^{h \times w}$), we use a standard cross-entropy plus soft Dice loss with $\lambda_{\text{dice}}=1$ and $\varepsilon = 1$, which are presented as: $\mathcal{L}_{\text{sup}} = \frac{1}{|\mathcal{D}_L|} \sum_{(x,y)} \left[ \text{CE}(y, p_S) + \lambda_{\text{dice}} \, \text{DSC}(y, p_S) \right]$, $\text{DSC}(y, p_S) = 1 - \frac{2 \sum_i y(i) \, p_S(i) + \varepsilon}{\sum_i y(i) + \sum_i p_S(i) + \varepsilon}$.

On unlabeled data, CPS (Sec. 3.2) provides *accepted* pseudo-labels using confidence and TTA-stability, SAT (Sec. 3.3) shapes local features near the boundary using agreement-gated positives, and PCM (Sec. 3.4) enforces a global head/background structure via uncertainty-weighted prototypes and a curriculum. The final training objective is a weighted sum: $\mathcal{L} = \mathcal{L}_{\text{sup}} + \lambda_{\text{CPS}} \, \mathcal{L}_{\text{CPS}} + \lambda_{\text{SAT}} \, \mathcal{L}_{\text{SAT}} + \lambda_{\text{PCM}} \, \mathcal{L}_{\text{PCM}}$ with defaults $\lambda_{\text{CPS}}=1.0$, $\lambda_{\text{SAT}}=0.3$, $\lambda_{\text{PCM}}=0.2$. Gradients flow through the *student* only and the teacher is updated by EMA (Eq. (1)). TTA averages are used for *gating* but are not part of the backpropagation graph. We follow a simple loop that keeps the CPS acceptance, SAT pairing, and PCM prototypes *consistent* by basing all decisions on the calibrated teacher probabilities from Sec. 3.2. We also prvovide the complete algorithm givenn above which shows the overall flow across different modules. Furthermore, we use AdamW (lr $1 \times 10^{-4}$, weight decay $1 \times 10^{-4}$), cosine decay with linear warmup (5% of steps), gradient clipping at 1.0, and set $\alpha=0.996$ for EMA.

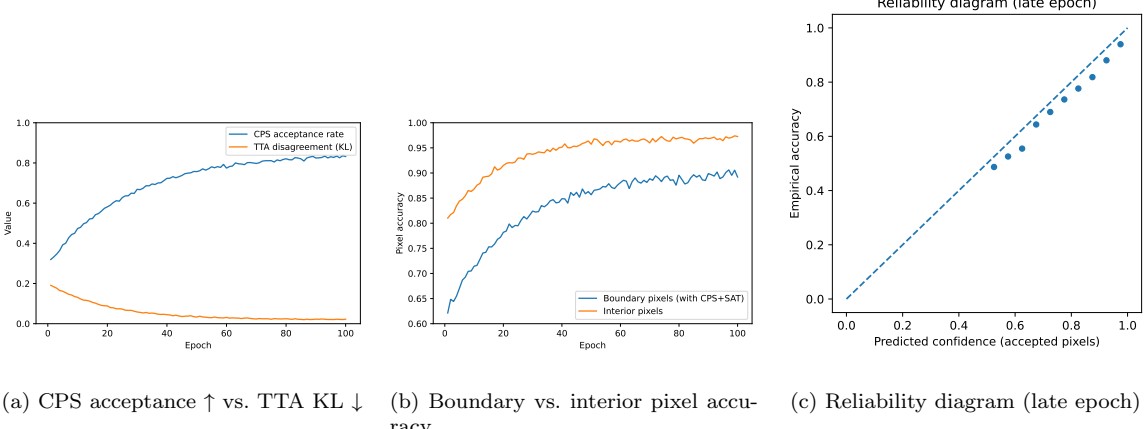

(a) CPS acceptance ↑ vs. TTA KL ↓    (b) Boundary vs. interior pixel accuracy    (c) Reliability diagram (late epoch)

Figure 4: Acceptance/uncertainty trends, boundary vs. interior accuracy, and calibration.

---

**Algorithm 1** UltraSemiNet Training Framework

---

**Input:** Labeled set $\mathcal{D}_L$, Unlabeled set $\mathcal{D}_U$; Student $\theta_S$, Teacher $\theta_T$; Calibration set $\mathcal{D}_{cal} \subset \mathcal{D}_L$.
**Output:** Optimized Student Model $\theta_S$
**Initialize:** Prototype EMA $\rho$, Teacher EMA $\alpha$, SAT radii $r$, Thresholds $\tau, \delta, \epsilon_1, \epsilon_2$. Warm-up prototypes $\mu_0, \mu_1$ using $\mathcal{D}_L$. **for** *epoch = 1 to E* **do**

    // 1.  Reliability Calibration (Once per epoch)
    Extract logits from $\mathcal{D}_{cal}$ using $\theta_T$. Minimize NLL to find scaling factor $\mathcal{T}$.
    **foreach** *batch* $\mathcal{B}_L \subset \mathcal{D}_L, \mathcal{B}_U \subset \mathcal{D}_U$ **do**
        // 2.  Multi-View Forward Pass
        Generat weak $(x^w)$ and strong $(x^s)$ views for $x \in \mathcal{B}_U$. Teacher : $p_T = \sigma(f(x^w; \theta_T)/\mathcal{T})$;
         Student : $p_S, z_S = f(x^s; \theta_S)$ (logits/feats).
        // 3.  CPS: Dual-Gating Mechanism
        Confidence: $m_{conf} = \mathbb{1}[\max(p_T) \geq \tau]$. Stability: $m_{stab} = \mathbb{1}[D_{KL}(p_T \| p_{TTA}) \leq \delta]$. Global
         Gate: $\mathcal{G} = m_{conf} \cdot m_{stab}$. Pseudo-label $\hat{y} = \text{argmax}(p_T)$. $\mathcal{L}_{CPS} = \frac{1}{|\mathcal{G}|} \sum \mathcal{G} \cdot \ell_{ce}(p_S, \hat{y})$.
          // 4.  SAT: Spatial Awareness & Boundary Belt
        Entropy map $H(p)$; Boundary Belt $\mathcal{B} = \mathbb{1}[\epsilon_1 \leq H(p) \leq \epsilon_2]$. Sample anchors $a$. If $a \in \mathcal{B}$,
         filter by structural consistency (sIoU $> \tau_b$). Contrastive loss $\mathcal{L}_{SAT}$ between anchors and
         context-aware negatives.
         // 5.  PCM: Prototype Consistency
        Update global prototypes $\mu_c \leftarrow \rho\mu_c + (1-\rho)\bar{z}_{\hat{y}}$ using confident features. Compute hardness
         $\kappa = p_S \cdot \text{sim}(z_S, \mu_{1-\hat{y}})$. Select top-ranked hard pixels. $\mathcal{L}_{PCM} = \ell_{contrast}(z_S, \mu_{\hat{y}}, \mu_{1-\hat{y}})$.
         // 6.  Optimization
        $\mathcal{L}_{total} = \mathcal{L}_{sup}(\mathcal{B}_L) + \lambda_1 \mathcal{L}_{CPS} + \lambda_2 \mathcal{L}_{SAT} + \lambda_3 \mathcal{L}_{PCM}$. Update $\theta_S \leftarrow \theta_S - \eta\nabla\mathcal{L}_{total}$; Update
        $\theta_T \leftarrow \alpha\theta_T + (1-\alpha)\theta_S$.
    **end**
**end**

---

Table 4: Key hyperparameters used in UltraSemiNet. We report defaults and suggested sweep ranges used in ablations.

| Name | Role | Default | Sweep |
|---|---|---|---|
| $r$ | Positive radius for SAT | 5 px | $\{3, 5, 7\}$ |
| $s$ | Local window size for $\mathcal{M}_i$ | 15 | $\{11, 15, 19\}$ |
| $[\epsilon_1, \epsilon_2]$ | Entropy belt thresholds | $[0.4, 0.95]$ | $\{[0.2, 0.9], [0.4, 0.95]\}$ |
| $\tau_{\mathrm{bIoU}}$ | Soft-IoU gate for boundary positives | 0.6 | $\{0.5, 0.6, 0.7\}$ |
| $q_b(t)$ | Boundary-positive admission (max) | 0.4 | $\{0.2, 0.4\}$ |
| $\gamma_1, \gamma_2$ | Anchor weight coefficients | 0.7, 0.3 | fixed |
| $\tau_c(t)$ | CPS class-wise thresholds | $0.95 \rightarrow 0.80$ | slope $\{\times 0.5, \times 1\}$ |
| $\delta$ | TTA disagreement (KL) cutoff | 0.15 | $\{0.10, 0.15, 0.20\}$ |
| $\tau$ | Temperature in $\pi_c(i)$ | 0.5 | $\{0.3, 0.5, 0.7\}$ |
| $m$ | PCM margin | 0.2 | $\{0.1, 0.2, 0.3\}$ |
| $\gamma(t)$ | Curriculum keep ratio (max) | 0.7 | $\{0.5, 0.7\}$ |

