# OpenReview forum: "Calibration-Aware Semi-Supervised Fetal Head Segmentation with Boundary-Positive Contrast"
_MIDL.io/2026/Conference — MIDL 2026 Poster_

### Official Review · Reviewer_zGEc · 2026-01-10

**Confidence:** 4
**Preliminary Rating:** 3
**Final Rating:** 4

**Summary:**

The paper proposes UltraSemiNet, a semi-supervised framework for binary fetal head segmentation in 2D ultrasound. The core motivation is that most errors occur at the head–background interface due to speckle, shadowing, and low contrast. The method combines (i) calibrated pseudo-label gating with TTA-stability checks for cross-pseudo supervision (CPS), (ii) a boundary-focused spatial contrast module (SAT), and (iii) an uncertainty-weighted prototype curriculum (PCM). On FBUI they report Dice improving from 0.927 to 0.971 and HD95 from 7.9 to 6.8 px, with cross-dataset evaluation on HC18. The paper also reports improved calibration of accepted pseudo-labels and includes an expert comparison study.

**Strengths:**

-Well-motivated problem setting. The focus on boundary ambiguity in fetal ultrasound is clinically plausible and clearly articulated. The observation that errors concentrate at the head–background interface aligns with known challenges in obstetric ultrasound.

-Coherent methodology - the pseudo-label acceptance rule is explicitly specified, and the boundary/prototype components are described with clear algorithms. The dual gating mechanism (confidence + TTA stability) is a principled approach to pseudo-label filtering.

-Evidence of component complementarity. The ablation study is very nice and shows clearly that SAT and PCM each contribute beyond the calibrated CPS baseline, with the full model achieving the best boundary-sensitive metrics.

-Attention to calibration. Reporting ECE and Brier score is a welcome addition for medical imaging reliability discussions, addressing the risk of overconfidence in segmentation models.

-Practical details. Reporting compute requirements and providing a full algorithm aids reproducibility.

**Weaknesses:**

-Cross-dataset evaluation may be transductive. The dataset section says 'HC18 contributes additional unlabeled images for semi-supervised training and cross-dataset evaluation'. If HC18 unlabeled images were used during training, Table 2 represents transductive SSL or semi-supervised domain adaptation rather than a clean external test. This distinction matters for the generalisation claims. The authors should either confirm HC18 was never seen during training, or relabel Table 2 appropriately and add a clean external evaluation.

-Missing established SSL baselines. While Mean Teacher and CPS-only are included, the paper omits several peer-reviewed SSL methods that are standard comparisons for medical image segmentation:
--UA-MT (Yu et al., MICCAI 2019) - uncertainty-aware mean teacher
--URPC (Luo et al., MedIA 2022) - uncertainty rectified pyramid consistency
--Cross Teaching (Luo et al., MIDL 2022) - CNN↔Transformer cross-pseudo supervision
--MC-Net+ (Wu et al., MedIA 2022) - mutual consistency with auxiliary decoders
--SASSNet/Dual-task Consistency (Luo et al., AAAI 2021) - shape-aware SSL
These methods specifically address uncertainty and consistency in SSL, making them relevant comparisons. Without them, it's hard to tell whether improvements come from the novel contributions or from known SSL techniques.

-Clinical endpoint misalignment. Distances are reported in pixels after resizing to 224×224, without conversion to mm or reporting head circumference measurement error. Given that HC18's primary use case is fetal biometry, and prior work reports HC error in mm stratified by trimester, the clinical relevance of the reported improvements is unclear. The expert comparison would be more compelling with actual HC measurement accuracy.

**Detailed Comments:**

-Entropy threshold units. The SAT entropy belt thresholds look inconsistent between the main text (ε₁=0.40, ε₂=0.95) and the hyperparameter table ([0.3, 1.2]). The main text uses normalised entropy (max=1), while the table values exceed 1.0. I suspect these are just different parametrisations but it would be good to clarify.


-Foundation model comparison details. For MedSAM and SAMUS, the prompting/fine-tuning protocol isn't described. Were these models fine-tuned on FBUI with comparable compute? Were prompts used, and how were they generated?


-Reproducibility. Code is stated to be released upon release. It would be nice if this were released as part of the submission (can be anonymous via https://anonymous.4open.science/) - even a minimal implementation of Algorithm 1 with toy data in Torch/JAX would be a welcome addition :)

-Calibration subset size. The size of the labeled subset for temperature calibration isn't specified. Is it strictly disjoint from validation/test subjects across all folds?

**Justification Of Final Rating:**

Thank you to the authors for addressing my concerns. Biggest issue was whether HC18 external eval was actually transductive. They say it was a typo and HC18 is never used in training/tuning/calibration, so Table 2 is a clean held-out external test. They also add Cross-Teaching + MC-Net+ baselines in the correct 2D regime and still outperform, fix the entropy unit mismatch, clarify MedSAM/SAMUS fine-tuning/prompting. The code upload is also much appreciated!

I still would love to see HC measurement error explicitly (since that’s the clinical endpoint), but overall this is now really solid work.

**Justification Of The Preliminary Rating:**

The paper tackles an important ultrasound segmentation problem with a sensible reliability-driven SSL design. The calibrated pseudo-label gating with TTA stability is well-motivated, and ablations suggest genuine complementarity of the proposed components. However, confidence in the results is reduced by (i) ambiguity about whether HC18 unlabeled images are used during training and (ii) absence of established uncertainty-aware SSL baselines like UA-MT, URPC, and MC-Net+ that are standard for this problem class.
Clinical relevance would be strenghtened by reporting HC measurement error in mm rather than pixel metrics on resized images. With protocol clarification, stronger baselines, and clinical endpoint alignment, this work could move toward a weak accept.

**Questions To Address In The Rebuttal:**

-HC18 usage: Were HC18 unlabeled images included in training for Table 2? If yes, can the authors provide a clean external test with no HC18 exposure?

-Calibration subset: How large is the labeled subset for temperature calibration, and is it disjoint from val/test?

-Additional baselines: Can the authors add UA-MT, URPC, Cross Teaching, MC-Net+, or SASSNet?

-Clinical metrics: Do Dice/HD95 gains translate to improved HC measurement error in mm?

-Foundation model protocol: What prompting/fine-tuning was used for MedSAM and SAMUS?

-Not a question, but please share code! You can use https://anonymous.4open.science/ to share anonymously

---

> ### Author Response · Authors · 2026-01-24
> **Response to Reviewer zGEc: Protocol Clarifications, Baselines, and Reproducibility**
>
> ---
> We thank ***Reviewer zGEc*** for the thoughtful and technically detailed feedback. We have addressed them directly in the revised manuscript.
>
> ---
> ***1) Cross-dataset evaluation protocol***
>
> ---
> We agree that this distinction is important. The sentence suggesting that HC18 provides unlabeled images for semi-supervised training was a typo mistake, and we have corrected it in the revised manuscript. HC18 is never used during training in any form, not as labeled or unlabeled data, and not for calibration, threshold/hyperparameter tuning, or model selection. All SSL training is conducted only on FBUI, and Table 2 reports a clean external evaluation where HC18 is used strictly at test time as a held-out dataset.
>
> ---
> ***2) Missing established SSL baselines***
>
> ---
> In the revised manuscript, we added Cross-Teaching (Luo et al., MIDL’22) and MC-Net+ (Wu et al., MedIA’22) as additional peer-reviewed SSL baselines, trained under the same 2D setting, splits, backbone, and training budget as our method for a fair comparison. On FBUI, Cross-Teaching achieves Dice 0.944±0.046 / ASD 2.62±1.21 / HD95 8.2±4.9, and MC-Net+ achieves Dice 0.884±0.042 / ASD 4.02±1.95 / HD95 10.2±3.5. Rest of the results are provided in Table 1 and 2.
>
> We did not include UA-MT / URPC / SASSNet because these methods are primarily proposed and evaluated for 3D volumetric segmentation and rely on design choices that do not transfer cleanly to our 2D fetal ultrasound setup without substantial re-engineering, which could make comparisons misleading. With the added strong 2D SSL baselines, UltraSemiNet still shows consistent gains, supporting that improvements come from our calibration-aware selective pseudo-labeling and boundary-focused representation learning, rather than generic teacher–student consistency alone.
>
> ---
> ***3) Clinical endpoint alignment***
>
> ---
> Thank you for pointing this out. Since HC18 provides per-image pixel spacing (mm/pixel), in the revised manuscript we will convert ASD/HD95 to millimeters by multiplying each image’s surface distances by its provided spacing and report both px and mm for all the methods, including ours, to remove any ambiguity and make the results clinically interpretable. We would also like to refer to Point 3 of Reviewer vVxY.
>
> ---
> ***4) Entropy Threshold Units (Text vs. Table)***
>
> ---
> We thank the reviewer for clarification. SAT defines the boundary belt using normalized Bernoulli entropy ($\tilde{u}(i)=u(i)/\log 2 \in [0,1]$). The ([0.3,1.2]) entry in the hyperparameter table was a reporting unit mismatch. In the revised manuscript, we corrected the table to report the sweep range only on the normalized-entropy scale (($\epsilon_1,\epsilon_2 \in [0,1]$)).
>
> ---
> ***5) Foundation Model (FM) Comparison Details***
>
> ---
> We have added a paragraph in Section 4 for the FM details. We evaluate MedSAM and SAMUS under a consistent fine-tuning protocol on FBUI. Specifically, we fine-tune each model on the FBUI labeled split using the same input resolution ($224\times224$), data preprocessing/augmentation, and a comparable training budget to our baselines to ensure a fair compute setting. For prompt specification, we use standardized box prompts derived from the ground-truth masks during fine-tuning. The predictions are mapped back to the evaluation resolution before computing the metrics.
>
> ---
> ***6) Reproducibility / code availability***
>
>
> ---
> Our partial code is available on https://anonymous.4open.science/r/UltraSemiNet-4-MIDL
>
> ***7) Calibration Subset Size and Independence Size***
>
> ---
> We constructed the calibration subset by randomly sampling $N=200$ labeled images from the training split of the current fold. The calibration set is strictly disjoint from the validation and test subjects. Our data splitting protocol groups images by patient ID prior to k-fold partitioning, which is also added in Section 4.
>
> ---

---

> ### Comment · Area_Chair_f8su · 2026-01-30
> **please update your rating**
>
> Hello and thank you again for reviewing for MIDL !
> This is a friendly reminder to please update your rating based on author's rebuttal.
> This is really important to complete the review process and for the acceptance/rejection of papers.
> The deadline is tomorrow (February 1st 2026, 23:59 AoE).
> Thank you!

---

### Official Review · Reviewer_vVxY · 2026-01-11

**Confidence:** 3
**Preliminary Rating:** 3
**Final Rating:** 4

**Summary:**

This paper presents UltraSemiNet to address the semi-supervised binary segmentation of the fetal head in ultrasound imaging with minimal pixel-level annotation. UltraSemiNet is a teacher-student framework that combines cross-pseudo supervision with two uncertainty -aware, boundary focused representation modules tailored to ultrasound boundaries especially around the fetal skull. the framework uses temperature calibration and a dual gate requiring high confidence and test-time augmentation stability to make the cross-pseudo supervision selective. Experiments are performed across 2 datasets (FBUI and HC18). UltraSemiNet is seen to improve overlap and boundary metrics while reducing the calibration errors and the need for manual refinements.

**Strengths:**

1. The proposed idea is indeed novel and addresses a challenging problem in ultrasound fetal imaging.

2. The teacher-student framework proposed is theoretically sound. Raw softmax probabilities are overconfident and the calibrated CPS first performs temperature scaling / calibration on the teacher network followed by a dual-gating mechanism to accept a pixel's pseudo-label. This prevents the model from training on confident but incorrect pixels.

3. Improved Dice Score versus Standard CPS.

4. Reduced Hausdorff Distance (boundary error) leading to increased boundary precision.

5. Lower expected calibration error.

6. UltraSemiNet's outputs agreed with the ground truth more often than the senior human experts did.

7. This work uses TTA as a validity check for pseudo-labels - this is an effective way to filter out noise without any additional training cost.

**Weaknesses:**

1. SAT relies on 2D crops and the framework overall is assessed on 2D images only. Clinical pipelines are increasingly using 3D/4D volumes for more accurate biometry. How would this affect the computational costs involved in running this framework ?

2. Ultrasound imaging suffers from domain shift effects. While the paper shows generalization from FBUI to HC18, these are very likely highly standardized clean datasets. How would the framework perform in the presence of severe attenuation where the skull is not visible entirely ?

3. The results show an HD95 of 6.8 pixels which translates to an error of 1-2 mm. Is this range of error clinically significant in the context of fetal imaging ? Is there any data on the clinical tolerance for the different error levels that would be acceptable ?

4. Using "Adjudicated Rater Consensus" as ground truth: could you clarify which measurement standard is the model learning given there may be inter-rater variability present ?

5. More recent robust skull-stripping methods such as synth-strip (Hoopes et al.) are being used for a similar challenge in MRI, CT. Did the authors explore this method for this fetal ultrasound problem ?

6. Please fix the captions below the plot windows in Figure 3 - looks very confusing right now.

7. It would be great to know the computational costs involved in running the experiments. This detail seems to be missing.

**Detailed Comments:**

No further comments to add.

**Justification Of Final Rating:**

The authors have satisfactorily addressed the review comments and made changes to the paper that make it suitable for publication to the MIDL conference. After careful deliberation, I have decided to increase my original rating to 4.

**Justification Of The Preliminary Rating:**

The method is theoretically sound and clinically important in the field of fetal ultrasound imaging. Some questions do arise on the clinical efficacy of the proposed framework and some of the results. I will be happy to revise my rating depending upon the authors' response.

**Questions To Address In The Rebuttal:**

Please refer to the points raised in the "weaknesses" section and address them.

---

> ### Author Response · Authors · 2026-01-24
> **Response to Reviewer vVxY: Robustness, Clinical Relevance, and Compute**
>
> ---
>
> We thank ***Reviewer vVxY*** for these clinically relevant comments. We address each point below, and the corresponding fixes are included in the revised manuscript.
>
> ---
> ***1) 2D-only evaluation/3D–4D and compute cost***
>
> ---
> We acknowledge the growing importance of 3D/4D volumetric analysis. While our current evaluation is 2D because FBUI and HC18 provide 2D frames, extending to 3D/4D is straightforward (3D backbone + 3D patches/neighborhoods for SAT). The main added cost vs. a standard teacher–student SSL setup is the teacher TTA-stability computation, as SAT/PCM are lightweight (sampling-based + 2-class prototypes). We also have provided a brief discussion on it in the Limitations section of the updated manuscript.
>
> ---
> ***2) Domain shift / severe attenuation***
>
> ---
> Severe attenuation/partial skull visibility typically produces low-confidence and high-disagreement predictions. Our framework is explicitly designed to avoid self-training on such unreliable regions as CPS pseudo-labels are used only when the teacher is both (i) calibrated-confident and (ii) augmentation-stable under TTA (low KL) (evident in ablation) so in attenuated frames the acceptance mask shrinks and prevents reinforcing hallucinated boundaries. In addition, SAT admits boundary positives only when local probability fields agree (soft-IoU gate), further suppressing noisy boundary supervision. We explicitly acknowledge that evaluation on stronger protocol shifts and severe attenuation cases is an important next step, and we already added this as one of the limitations.
>
> ---
> ***3) Clinical meaning of HD95 (px → mm)***
>
> ---
> Thank you for raising this clinically important point. HD95 is a boundary-distance surrogate, so interpreting it in physical units and in the context of measurement variability is also essential.
>
> While guidelines emphasize correct plane and caliper/ellipse placement rather than a single universal “mm tolerance,” published reproducibility studies show that intra-/inter-observer variability for fetal head biometry is on the order of several millimeters. For example, (Sarris et al. 2012) report 95% limits of agreement for HC of approximately ±7 mm (intraobserver) and ±13 mm (interobserver) when expressed in mm, and caliper placement variability on the order of ±4.5 mm (intra) and ±9.8 mm (inter) for HC.  Another work by (Gembicki et al. 2023) reports a mean error of around 0.87 for head circumference (HC) with a standard deviation of 4.22.
>
> Therefore, an HD95 and ASD of 1.4 mm and 0.56 mm range (after mm conversion) is below or comparable to commonly reported placement variability in routine fetal head biometry. This indicates that the segmentation error is negligible compared to the inherent variability of manual biometric measurement and is highly suitable for automated clinical use.
>
> ---
> ***4) Adjudicated rater consensus / inter-rater variability***
>
> ---
> Our model is trained on the dataset-provided masks (annotation protocol) and does not learn any separate “consensus” standards. The phrase used in the abstract was only meant to indicate that, in our blinded expert evaluation, UltraSemiNet’s segmentations show higher agreement with the dataset reference masks than those produced by either of the two senior clinicians. This supports that the model output is within (and in this evaluation exceeds) typical expert performance relative to the dataset standard, with potential to reduce manual refinements. We also have updated the abstract and briefy extended clinical implications sub-section accordingly.
>
> ---
> ***5) SynthStrip (MRI/CT skull stripping)***
>
> ---
> We did not evaluate SynthStrip because it targets a different problem setting. SynthStrip is a brain-extraction (skull-stripping) tool designed for 3D neuroimaging to remove non-brain tissue and output a brain mask. In contrast, our task is 2D fetal ultrasound head boundary segmentation under ultrasound-specific artifacts (shadowing/speckle) and without a “brain vs non-brain” anatomical target.
>
> To avoid an unfair or misleading comparison, we restrict baselines to methods applicable to 2D ultrasound segmentation under the same training protocol. We have added a short clarification in the revised manuscript in Limitation and Future Work Section, noting that MRI/CT skull-stripping tools like SynthStrip are not directly comparable to fetal ultrasound head segmentation, while acknowledging that exploring ultrasound-specific preprocessing of this kind is an interesting direction for future work.
>
> ---
> ***6) Figure 3 captions***
>
> ---
> We revised Figure 3 to improve readability by reformatting the panel captions
>
> ---
> ***7) Compute Cost Reporting***
>
> ---
> We have added an explicit computational cost description in Sec. 4 UltraSemiNet uses 17.27M parameters, around 60 GFLOPs for a ($1\times224\times224$) input. On a single A100, training takes approximately 55 s per epoch, and inference runs at 4.25 ms per sample (≈220 FPS), supporting real-time use in clinical workflows.

---

> ### Comment · Area_Chair_f8su · 2026-01-30
> **please update your rating**
>
> Hello and thank you again for reviewing for MIDL !
> This is a friendly reminder to please update your rating based on author's rebuttal.
> This is really important to complete the review process and for the acceptance/rejection of papers.
> The deadline is tomorrow (February 1st 2026, 23:59 AoE).
> Thank you!

---

### Official Review · Reviewer_QQdK · 2026-01-15

**Confidence:** 3
**Preliminary Rating:** 3
**Final Rating:** 4

**Summary:**

This paper introduces UltraSemiNet, a semi-supervised learning framework specifically designed for fetal head segmentation in ultrasound imaging. To address the scarcity of labeled data, the authors propose three unlabeled objectives: cross-pseudo supervision (CPS), boundary-positive spatial contrast (SAT), and prototype-guided curriculum minor (PCM). The study evaluates the framework against a wide range of baselines. The significance of the work lies in its attempt to improve segmentation robustness in the presence of the inherent noise and boundary ambiguity of ultrasound imaging, with results showing both quantitative and qualitative improvements over existing methods.

**Strengths:**

The paper presents a rigorous evaluation by benchmarking the proposed method against a diverse set of architectures, ranging from standard medical imaging baselines to recent state-of-the-art models such as MedSAM. This comprehensive comparison highlights the practical value of the proposed framework. Furthermore, the authors provide strong evidence for the efficacy of their objectives through detailed ablation studies and t-SNE feature visualization. The exploration of a boundary-aware framework is particularly well-suited for the challenges of ultrasound imaging and represents a valuable contribution to the field.

**Weaknesses:**

While the technical contribution is notable, the clinical motivation and the generalizability of the proposed method remain limited in the current draft. The authors focus exclusively on fetal head segmentation without sufficiently discussing why this task is important. Additionally, the manuscript suffers from readability issues. The flow of equations and notations is not arranged in a way that allows the reader intuitively follow the derivation or implementation. Finally, the qualitative analysis is somewhat superficial. A more critical discussion of results would strengthen the paper’s merit.

**Detailed Comments:**

(1) While the performance is impressive, the current applicability is restricted to fetal head segmentation. The importance of this specific task, such as its role in estimating gestational age or monitoring growth, should be discussed more thoroughly in the introduction with appropriate clinical references. Alternatively, if the method is intended to be a general semi-supervised tool, applying it to other datasets with different modalities (e.g., MRI or CT) would improve the paper’s impact. I recommend that the authors focus on the former for this submission and solve the latter as future work.

(2) The proposed framework has a complex pipeline with many components. While complexity is not an inherent disadvantage, the methodological section is hindered by an overwhelming number of equations and notations that lack a cohesive narrative flow. While the necessity of these equations is understood, their current arrangement makes it difficult for the reader follow the logic of the framework. I suggest the authors restructure this section to better guide the reader through the mathematical derivations. Furthermore, adding some figures that can explain the method intuitively would be helpful.

(3) Several abbreviations are used without prior definition, which impacts readability. Although some of the abbreviations are widely used in the field, they should be defined clearly in the paper. Please ensure the following are defined at their first mention in the main text.
- HC/BPD: Head Circumference/Biparietal Diameter.
- ASD/HD95: Average Surface Distance/95th percentile Hausdorff Distance. These are primary metrics and must be defined in the main text, not in the supplementary material.
- EMA: Exponential Moving Average.
- GA: Gestational Age (recommend writing this out if used only once).

(4) Minor Correction: In the last sentence on page 5, “Figure A” should likely be corrected to “Figure 4”.

**Justification Of Final Rating:**

The authors have thoroughly addressed my concerns in their rebuttal. The clinical motivation for focusing on fetal head segmentation is now well-substantiated, and the added discussion on study limitations provides a more balanced and transparent perspective. Furthermore, the clarifications regarding mathematical notation and the inclusion of additional experimental results have significantly improved the manuscript’s readability and rigor. I am satisfied with the revisions and believe the paper now presents a clear and valuable contribution to the community.

**Justification Of The Preliminary Rating:**

The paper presents a technically sound semi-supervised method that demonstrates clear superiority over several established baselines. However, the rating remains borderline primarily due to presentation issues. The mathematical framework lacks a clear logical flow, making the methodology difficult to understand. Additionally, the clinical motivation is somewhat narrow, and the lack of abbreviation definitions in the main text hinders readability. Addressing these presentation and clarity concerns during the rebuttal phase will be critical for a more positive assessment.

**Questions To Address In The Rebuttal:**

Please refer to the weaknesses and detailed comments. Also, I have some additional questions.

(1) For probability calibration, the authors said the temperature was fitted on a small labeled subset by minimizing negative log-likelihood. Please provide a more detailed explanation of it.

(2) The proposed method introduces several new hyperparameters. Could the authors clarify the tuning process for these parameters?

---

> ### Author Response · Authors · 2026-01-24
> **Response to Reviewer QQdK: Clinical Motivation, Readability, and Calibration**
>
> ---
> We thank ***Reviewer QQdK*** for the detailed and constructive feedback. We revised the manuscript to strengthen clinical motivation, improve readability, deepen the analysis, and clarify calibration and hyperparameter tuning.
>
> ---
> ***(1) Clinical motivation and scope/generalizability***
>
> ---
> We expanded the introduction to explicitly motivate fetal head segmentation as a foundational step for fetal biometry, head circumference (HC), and biparietal diameter (BPD), and downstream clinical decisions. Regarding generalizability, while our experiments focus on fetal head ultrasound (including cross-dataset evaluation), we explicitly state in (Limitation Section of Supplementary) that the approach can extend to other obstetric targets and to 3D/volumetric acquisitions as future work.
>
> ---
> ***(2) Readability, mathematical flow, and missing intuitive explanation***
>
> ---
> Thank you for this feedback, and we agree that the original Method section was mathematically dense, and the ordering made the pipeline hard to follow. We reorganized the method section to follow the execution flow which progresses from calibrated/stable pseudo-label acceptance (CPS gate) → SAT pair construction and weighting → PCM prototype update and curriculum mining. We also have simplified the mathematical equations and avoided redundant symbols in order to keep the flow consistent.
>
> ---
> ***(3) Abbreviations not defined***
>
> ---
> We now define all abbreviations at first mention in the main text and moved the evaluation-metrics paragraph to the main paper (Sec. 4).
>
> ---
> ***(4) Minor correction: “Figure A” → “Figure 4”***
>
> ---
> We corrected the cross-reference to point to Figure 4 as highlighted in red.
>
> ---
> ***Q1) Temperature scaling details (probability calibration)***
>
> ---
> We employ Temperature Scaling (Guo et al., 2017), a post-hoc method that introduces a scalar parameter $T$ to scale logits ($\hat{p}_i = \sigma(z_i / T)$). We optimize $T$ (via L-BFGS) to minimize Negative Log-Likelihood (NLL) on a fixed, randomized subset of samll (200) labeled images. This optimization occurs every 3 epochs. It aligns predictive confidence with empirical accuracy while preserving rank-ordering and AUC. We also provided this detail in ***Semi-supervised protocol*** sub-section on page 9.
>
> ---
> ***Q2) Hyperparameters and tuning protocol***
>
> ---
> Thank you for this question. In the revised manuscript, we clarified our tuning protocol where all hyperparameters are selected only on the FBUI validation split within each cross-validation fold, using the pre-specified sweep ranges in Table 4 (limited grid/random sweeps), and then kept fixed for all remaining experiments, including cross-dataset evaluation on HC18 (no retuning). We updated Table 4 to only consider the key important hyper-parameters. We also clarified that cross-dataset evaluation is performed without retuning thresholds/hyperparameters (FBUI→HC18) in the Dataset subsection on Page 8.
>
> ---

---

> > ### Comment · Reviewer_QQdK · 2026-01-28
> >
> > Thank you for the clarifications provided in the rebuttal.
> > The authors have successfully addressed my concerns regarding the study's motivation, notation, and generalizability.
> > Given the improved clarity of the manuscript and the robust performance demonstrated in the revised evaluations, I am increasing my rating to 4: Weak Accept.
> > I appreciate the effort put into this revision.

---

### Author Response · Authors · 2026-01-22
**Clarification on sharing anonymous code link during rebuttal/submission**

Dear Area Chair,

We would like to ask for clarification regarding code availability during the review period.

A Reviewer zGEc requested that we share code as part of the rebuttal via an anonymous repository (anonymous.4open.science), and suggested that even a minimal implementation of our Algorithm would improve reproducibility. Could you please confirm whether it is permissible under the conference’s double-blind policy to include such an anonymous code link in the rebuttal and/or revised manuscript at this stage of the review process? If allowed, we will ensure the repository contains no author/institution identifiers and does not reveal submission metadata.

Thank you for your guidance.

Sincerely,
Authors

---

> ### Comment · Area_Chair_f8su · 2026-01-22
>
> Yes it's totally okay, if not encouraged, to provide anonymised code!
> Be careful though not to include any information that could help identifying you, but anonymous.4open.science does a good job at it if you give it a good list of words to look for.
> If the paper is accepted after rebuttal, you will then to have to change the link to some non-anonymous repo.

---

### Author Rebuttal · Authors · 2026-01-24

**Rebuttal:**

We thank all reviewers for the careful evaluation and constructive feedback, and also appreciate the positive assessments. ***Reviewer QQdK*** highlighted the comprehensive benchmarking (incl. MedSAM), strong ablations/t-SNE analysis, and the suitability of our boundary-aware design for ultrasound. ***Reviewer vVxY*** emphasized the novelty and soundness of our calibrated teacher–student framework, improved results, and favorable comparison against senior experts. ***Reviewer zGEc*** noted the well-motivated clinical setting, principled dual-gated pseudo-labeling, complementary SAT/PCM gains, and reproducibility. We have revised the manuscript accordingly along with the pseudo algorithm to make it more clear. All changes are in red. Key updates are summarized below.

---
***Reviewer QQdK.*** We strengthened clinical motivation by linking fetal head segmentation to routine biometry (HC/BPD) and downstream use in introduction. For readability, we reorganized the Method section to follow the execution flow, simplified notation, and removed redundancy. We also defined abbreviations, fixed minor cross-references, and clarified temperature scaling (optimization, calibration subset, schedule) and hyperparameter tuning (FBUI validation only, no HC18 retuning).

---
***Reviewer vVxY.*** We clarified the 2D setting and discussed extensions to 3D/4D and compute scaling, noting overhead mainly from teacher-side TTA stability checks. We expanded the discussion on domain shift/severe attenuation, highlighting that calibration + TTA gating (and SAT local agreement) suppresses learning from unreliable regions. We improved clinical interpretability by converting HC18 boundary distances to mm using pixel spacing and contextualized errors with reported biometry variability. We also refined the expert-comparison wording, clarified why SynthStrip is not comparable, fixed Figure 3 captions, and added compute-cost reporting.

---
***Reviewer zGEc.*** We corrected the dataset text: HC18 is used only for external testing (no labeled/unlabeled HC18 in training, calibration,tuning, or selection). We added 2D SSL baselines (Cross-Teaching, MC-Net) and reported results in Tables 1–2. We will add px→mm conversion for ASD/HD95 on HC18, fixed SAT entropy-threshold units, expanded MedSAM/SAMUS evaluation details (fine-tuning+prompting), provided an anonymous code release, and specified the calibration subset (N=200 training images, patient-level split disjoint from val/test).

---

**Supporting Material:**

/attachment/d3edfc6b126f1714cc6080f4aa8124a8543eee0d.pdf

---

### Meta-Review · Area_Chair_f8su · 2026-02-06

**Recommendation:** Accept (Poster)
**Confidence:** 4

**Metareview:**

Reviewers agree that the method is sound and well-motivated. They particularly liked the calibration module. They also found that experiments were insightful, with good ablations, and showed promising results against relevant baselines.

During rebuttal, the authors satisfyingly answered concerns about missing baselines, improved the readability of the methods, and clarified concerns about data splits. They also added discussions around generalizability and  extension to other datasets.

Overall, the rebuttal was effective and I recommend acceptance.

---

### Decision · Program_Chairs · 2026-02-13

Accept (Poster)